# SBGC: Bidirectional Graph Comparison-Based Self-Supervised Network for Change Detection in Heterogeneous Images

## Abstract

Change detection (CD) in heterogeneous images is a hot but highly challenging topic in the field of remote sensing. However, the significant imaging differences and varying visual appearances of heterogeneous images complicate the accurate detection of changes occurring on the land surface through direct comparison. To overcome this challenge, this paper proposes a self-supervised network based on bidirectional graph comparison (SBGC) for unsupervised heterogeneous CD, which exploits modality-independent structural relationships. First, pseudo-Siamese networks are established to extract discriminative and robust features from bi-temporal heterogeneous images based on self-supervised contrastive learning. Then, these learned features are utilized to construct graph structures that represent structural relationships. Second, we introduce bidirectional graph comparison to fully exploit the graph structures for exploring comprehensive change information. Specifically, we map the graph structures to their opposite image modality and perform a bidirectional comparison between the original and mapped graph structures to generate a difference image. Finally, the change map is obtained by applying the Otsu segmentation algorithm to the difference image. Experimental results on three public heterogeneous datasets with different modality combinations show that the proposed method achieves superior performance compared to seven state-of-the-art methods, achieving the best performance with an average overall accuracy of 96.69%.

## 1 Introduction

Change detection (CD) utilizes remote sensing imagery captured at different times to analyze changes in ground objects within the same geographical area (Wen et al., 2021), which facilitates various applications, such as disaster assessment (Qing et al., 2022), and environmental monitoring (Kalinaki et al., 2023).

CD techniques can be categorized into two types based on the data sources of the images: homogeneous CD and heterogeneous CD. Homogeneous CD, which employs images captured by the same sensor, is an intuitive approach that has succeeded in various applications (Du et al., 2019; Zhan et al., 2023; Ma et al., 2023; Fang et al., 2024; Cui et al., 2024; Ding et al., 2024). However, its practicality is limited in some extreme scenarios, such as rapid responses to severe natural disasters, where timely images from the same sensor may not be available, rendering it ineffective for detecting changes in ground objects. In contrast, heterogeneous CD, which utilizes images from different sensors, can effectively overcome this limitation. In recent years, advancements in remote sensing technology have increased the availability of remote sensing images from different sources, providing valuable data support for heterogeneous CD (Liu et al., 2024). These images can be captured using different sensors, including synthetic aperture radar (SAR), optical, and hyperspectral sensors. Generally, detecting changes in ground objects between heterogeneous images is more challenging than between homogeneous images, because heterogeneous CD should consider not only the challenges such as noise and illumination that homogeneous CD faces, but also the modality discrepancy.

Heterogeneous CD, where pre-event and post-event images are captured by different sensors, has become popular due to the complementary strengths of heterogeneous images, thereby enhancing the response efficiency and accuracy of CD. Nonetheless, heterogeneous images exhibit distinct imaging mechanisms and varied depictions of the same land cover, rendering direct comparison through pixel/object difference measurement for CD impractical (Zhan et al., 2017; Caye Daudt et al., 2018). To address this challenge, researchers have been exploring the modality-independent structural relationships within heterogeneous images. Wan *et al.* (L. Wan & You, 2018) utilized sorted histograms to capture the local internal layout of the image and then computed the differences between the sorted bins to detect changes. Luppino *et al.* (Luppino et al., 2019) represented structural relationships by constructing local affinity matrices, subsequently calculating the differences between these matrices. In (Sun et al., 2021b; 2022), structural relationships were represented through the construction of nonlocal graph structures, and the structural differences between heterogeneous images were then computed using graph mapping. Sun *et al.* (Sun et al., 2021a) further introduced an iterative robust graph and Markovian co-segmentation framework (IRG-McS) that utilizes superpixels to represent ground object information, treating them as nodes for graph construction. In (Jimenez-Sierra et al., 2022), a framework based on nonlocal graph structures driven by signal smoothness representation was proposed. This nonlocal graph-based approach (Sun et al., 2021b; 2022; 2021a; Jimenez-Sierra et al., 2022) constructs nonlocal k-nearest neighbors (KNN) graphs and computes the similarity levels of these graph structures to detect changes. However, there are two main issues with this approach. First, this approach relies solely on original pixel information in images, which lacks robustness in complex scenes with diverse land cover types and varying sizes. Second, concerning mapping the graph structure from the pre-event image to the post-event image, for instance, the approach uses a one-way graph comparison, focusing solely on the comparison between the mapped and the post-event image's graph structure. However, it neglects the original pre-event graph structure and fails to fully utilize all the graph structure information, leading to more false detections.

To overcome the issues mentioned above, we propose a **S**elf-supervised network based on **B**idirectional **G**raph **C**omparison (SBGC) for unsupervised heterogeneous CD. First, pseudo-Siamese networks are established for self-supervised learning (SSL), aiming to extract representative and robust features directly from the original images. Second, to fully exploit the change information leveraging the graph structures, we propose bidirectional graph comparison (BGC). Specifically, we map the graph structures into their opposite image modality and subsequently conduct the bidirectional comparison between the original and mapped graph structures. The main contributions of this work are summarized as follows.

1) Efficient pseudo-Siamese networks are employed to learn robust and representative features from original images through SSL, effectively handling complex scenes with varied land cover and sizes, thus improving the accuracy of CD.

2) We propose BGC to fully explore the rich information within the graph structures. Specifically, we perform a bidirectional comparison between the original and mapped graph structures to extract difference information.

3) The impressive experimental results on three public heterogeneous datasets with different modality combinations demonstrate the superiority and practicality of SBGC in comparison with seven state-of-the-art (SOTA) unsupervised heterogeneous CD methods.

The rest of this article is structured as follows. Section 2 presents the proposed method in detail, followed by the experimental results and discussions in Section 3, and finally, the conclusion of our work is provided in Section 4.

## 2 METHODOLOGY

### 2.1 OVERALL FRAMEWORK

The framework of the proposed SBGC for unsupervised heterogeneous CD is shown in Fig. 1, consisting of three key steps: (1) image patches guided SSL; (2) graph construction and BGC; and (3) generation of the change map. We consider bi-temporal heterogeneous images, denoted as $X \in \mathbb{R}^{H \times W \times C_X}$ with modality $\mathcal{X}$ and $Y \in \mathbb{R}^{H \times W \times C_Y}$ with modality $\mathcal{Y}$ covering the same geographical

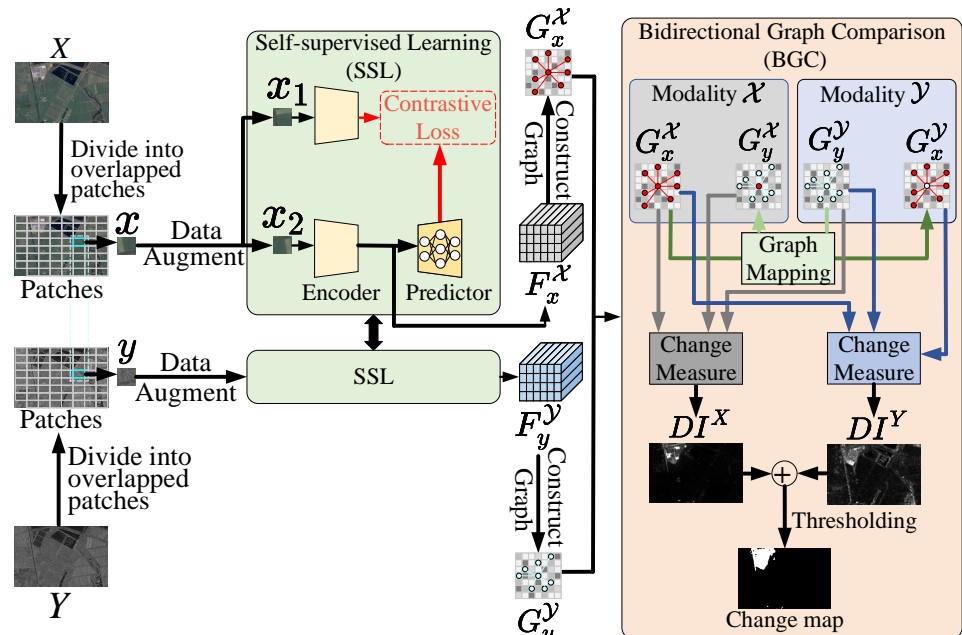

Figure 1: The framework of the proposed SBGC. (Here, for simplicity, we only present the SSL process associated with $X$, noting that the learning process associated with $Y$ is identical to that of $X$, with the only difference being the input channel dimension of the image patches $x$ and $y$)

area at times $t_1$ and $t_2$. Here $H$, $W$, and $C_X$ ($C_Y$) denote the height, width, and the corresponding channels. First, we divide $X$ and $Y$ into overlapping image patches to serve as our training samples. Subsequently, data augmentation is applied to these patches to generate two different views. The augmented patches are then input into pseudo-Siamese networks, which learn discriminative and robust features by optimizing a contrastive loss function. Next, the learned features are utilized to capture the structural relationship through graph construction. Thereafter, BGC is employed to the constructed graphs to fully exploit the graph information, resulting in $DI^X$ and $DI^Y$. Finally, $DI^X$ and $DI^Y$ are fused to generate the final difference image, which can be thresholded to obtain the change map.

## 2.2 IMAGE PATCHES GUIDED SSL

Following (Sun et al., 2022), both images are divided into overlapping square patches denoted as $x^i \in \mathbb{R}^{p \times p \times C_X}$ and $y^i \in \mathbb{R}^{p \times p \times C_Y}$ using a sliding window approach, where the patch size is set to $p$ with a step length of $\lceil p/2 \rceil$. Here, $i \in \{1, \ldots, M\}$, where $M$ represents the number of patches.

In the community of remote sensing, particularly for heterogeneous CD task, supervised feature learning usually requires sufficient labeled samples, which are difficult to obtain in real applications. In contrast, SSL is able to learn useful feature representations from raw data without manually labeling samples. Consequently, we employ SSL to extract more representative features from the original images, thereby enhancing the accuracy of CD by leveraging more representative information rather than relying solely on original pixel data. As depicted in Fig. 1, we build pseudo-Siamese networks for SSL, comprising two branches. The lower branch comprises one encoder, and one predictor, while the upper branch shares the same network architecture, excluding the predictor. Here, we use a sub-ResNet18 (He et al., 2016) as the encoder, modifying the stride of the first layer from 1 to 2 and removing the third and fourth layers to adapt to relatively small input patch size. The predictor is composed of fully connected layers with the structure 512-128-512.

During the SSL process, data augmentation is initially applied to $x^i$ to generate two distinct views: $x_1^i$ and $x_2^i$, which are then regarded as a positive sample pair. Here, we follow the reference augmentations in (Grill et al., 2020), including horizontal flipping and vertical flipping. Subsequently, $x_1^i$ and $x_2^i$ are input into the respective branches of the pseudo-Siamese network to obtain feature vec-

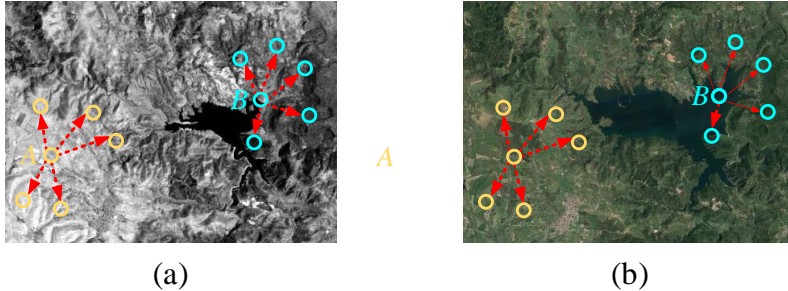

(a)                           (b)

Figure 2: Structural relationship in heterogeneous images. (a) Pre-event image. (b) Post-event image. The thickness of the dashed lines represents the similarity between regions, with thicker lines indicating greater similarity.

tors $z_{x^i}$ and $p_{x^i}$. A cosine similarity function $d_\theta(\cdot)$ is then employed to assess the similarity between the positive sample pairs. The pseudo-Siamese network associated with $X$ is trained to minimize the contrastive loss $\mathcal{L}_{CL}^X$, which is defined as follows:

$$\mathcal{L}_{CL}^X = -\sum_{i=1}^{M} \log \frac{d_\theta(z_{x^i}, p_{x^i})}{\sum_{j=1}^{M} d_\theta(z_{x^j}, p_{x^j})} \tag{1}$$

where positive sample pairs are assigned a higher value representing a close distance, thereby facilitating the extraction of discriminative features from the original images. Similarly, the above step can also be performed to optimize the network associated with $Y$ using the corresponding contrastive loss $\mathcal{L}_{CL}^Y$. Then the final loss can be written as

$$\mathcal{L}_{CL} = \frac{1}{2}(\mathcal{L}_{CL}^X + \mathcal{L}_{CL}^Y) \tag{2}$$

After the networks are trained, all the image patches from $X$ and $Y$ are fed into the respective encoder to obtain the deep feature representations, which can be denoted as $\{F_{x^1}^X, ..., F_{x^M}^X\}$. Similarly, we can get $\{F_{y^1}^Y, ..., F_{y^M}^Y\}$ from $Y$.

## 2.3   Graph Construction and BGC

In this work, we construct KNN graphs to capture the structural relationships for change measurement. For instance, as depicted in Fig. 2, the unchanged region $A$ maintains the KNN graph structure formed by $A$ and its similar regions, exhibiting minimal changes after the event. In contrast, the KNN graph structure formed by the changed region $B$ cannot maintain consistency due to the occurrence of changes. We define this relationship as the structural relationship, which is established through the construction of KNN graphs for the image regions.

For every patch $x^i$ from $X$, we construct its KNN graph $G_{x^i}^X = \{V_{x^i}^X, E_{x^i}^X, W_{x^i}^X\}, i = 1, 2, ..., M$ by finding its $K$ most similar patches based on their feature similarity. The graph structure is then represented as follows:

$$\begin{cases} \mathcal{V}_{x^i}^X = \{F_{x^i}^X, F_{x^k}^X | k = 1, 2, ..., K\}, |\mathcal{V}_{x^i}^X| = K+1 \\ \mathcal{E}_{x^i}^X = \{(F_{x^i}^X, F_{x^k}^X) | F_{x^k}^X \in \mathcal{V}_{x^i}^X\} \\ W_{x^i}^X = \{\text{dist}(F_{x^i}^X, F_{x^k}^X) | (F_{x^i}^X, F_{x^k}^X) \in \mathcal{E}_{x^i}^X\} \end{cases} \tag{3}$$

where $x^k$ denotes the $k$th patch most similar to $x^i$, $V_{x^i}^X$ represents the vertex set of the graph, $E_{x^i}^X$ denotes the edge set, and each edge connects two vertices. $W_{x^i}^X$ measures the weight of the edges, calculated using the formula $\text{dist}(F_{x^i}^X, F_{x^k}^X)$, where $\text{dist}(\cdot)$ denotes the squared Euclidean distance. Similarly, we can construct a graph $G_{y^i}^Y = \{\mathcal{V}_{y^i}^Y, \mathcal{E}_{y^i}^Y, W_{y^i}^Y\}$ of $y^i$. Following the approach in (Sun et al., 2021a), we set an adaptive $K$ as $\lceil (\sqrt{M} + \frac{\sqrt{M}}{10})/2 \rceil$.

Table 1: Description of the three heterogeneous datasets.

| Dataset | Sensor | Size | Location | Date | Change Event |
|---------|--------|------|----------|------|--------------|
| Sardinia | Landsat-5/Google Earth | $300 \times 412 \times 1(3)$ | Sardinia, Italy | Sep. 1995/Jul. 1996 | Lake expansion |
| Shuguang | Radarsat-2/Google Earth | $593 \times 921 \times 1(3)$ | Shuguang Village, China | Jun. 2008/Sep. 2012 | Building construction and river expansion |
| California | Landsat-8/Sentinel-1A | $875 \times 500 \times 11(3)$ | Sutter County, USA | Jan. 2017/Feb. 2017 | Flooding |

We can also construct a KNN graph $G_{x^i}^{\mathcal{Y}} = \{\mathcal{V}_{x^i}^{\mathcal{Y}}, \mathcal{E}_{x^i}^{\mathcal{Y}}, W_{x^i}^{\mathcal{Y}}\}$ to map $G_{x^i}^{X}$ from modality $X$ to modality $\mathcal{Y}$. This graph is constructed using the spatial coordinates of the $K$ patches most similar to $x^i$. Most methods (Sun et al., 2021b; 2022; Chen et al., 2022; Sun et al., 2021a) derive change information by comparing the differences between $G_{x^i}^{\mathcal{Y}}$ and $G_{y^i}^{\mathcal{Y}}$. However, these methods only perform one-way difference calculations, emphasizing changes in modality $\mathcal{Y}$ while neglecting the differences between $G_{x^i}^{\mathcal{Y}}$ and $G_{x^i}^{X}$, thus not fully utilizing all graph information. Therefore, we propose BGC to fully use the information from $G_{x^i}^{X}$, $G_{x^i}^{\mathcal{Y}}$, and $G_{y^i}^{\mathcal{Y}}$, resulting in richer change information and achieving better CD performance. Specifically, we calculate the differences of $(G_{x^i}^{X}, G_{x^i}^{Y})$ and $(G_{x^i}^{Y}, G_{y^i}^{\mathcal{Y}})$ for change measurement, obtaining difference information $d_i^X$ computed as

$$d_i^X = \frac{1}{K} \sum_{k=1}^{K} \left| ||F_{x^i}^X - F_{x^k}^X||_2^2 - ||F_{x^i}^{\mathcal{Y}} - F_{x^k}^{\mathcal{Y}}||_2^2 \right| + \\ \left| ||F_{x^i}^{\mathcal{Y}} - F_{x^k}^{\mathcal{Y}}||_2^2 - ||F_{y^i}^{\mathcal{Y}} - F_{y^k}^{\mathcal{Y}}||_2^2 \right| \tag{4}$$

The above step can also be performed to obtain $G_{y^i}^{X}$ for $G_{y^i}^{\mathcal{Y}}$. Similar to $d_{\Omega_i}^X$, we can obtain $d_i^{\mathcal{Y}}$. Additionally, we can derive $DI^X$ and $DI^Y$, denoted as

$$DI^X(h, w) = d_i^X \tag{5}$$

$$DI^Y(h, w) = d_i^{\mathcal{Y}} \tag{6}$$

where $(h, w)$ denote the spatial coordinates of patches $x_i$ and $y_i$, and $i = 1, 2, \ldots, |M|$.

### 2.4 GENERATION OF THE CHANGE MAP

The generated $DI^X$ and $DI^Y$ can only capture change information within their respective modalities, lacking the capability to account for the global change information. Therefore, these two difference images should be fused to produce a more robust difference image $DI^{\text{final}}$. Specifically, we utilize a direct averaging operation for this fusion, as follows:

$$DI^{\text{final}} = (DI^X/max(DI^X) + DI^Y/max(DI^Y))/2 \tag{7}$$

During the difference image analysis stage, the CD problem can be treated as an image segmentation problem, which can be solved by using threshold-based (Otsu, 1979; Moser & Serpico, 2006) or clustering-based (Krinidis & Chatzis, 2010; Gong et al., 2012) approaches employed in traditional homogeneous CD. Here, we directly utilize a simple threshold segmentation algorithm, Otsu (Otsu, 1979), to classify each pixel in $DI^{\text{final}}$ into changed class or unchanged class, which is given by

$$CM(i, j) = \begin{cases} 1, DI^{\text{final}}(i, j) > T \\ 0, DI^{\text{final}}(i, j) \leq T \end{cases} \tag{8}$$

where $T$ is a threshold value, $CM$ is the binary change map, in which the labels of changed and unchanged locations are labeled as "1" and "0", respectively.

## 3 EXPERIMENTS

### 3.1 DATASETS

To evaluate the effectiveness of SBGC, we conduct experiments on three public heterogeneous datasets, as shown in Fig. 3(a)-(c). The first dataset (Touati et al., 2020) was collected in Sardinia,

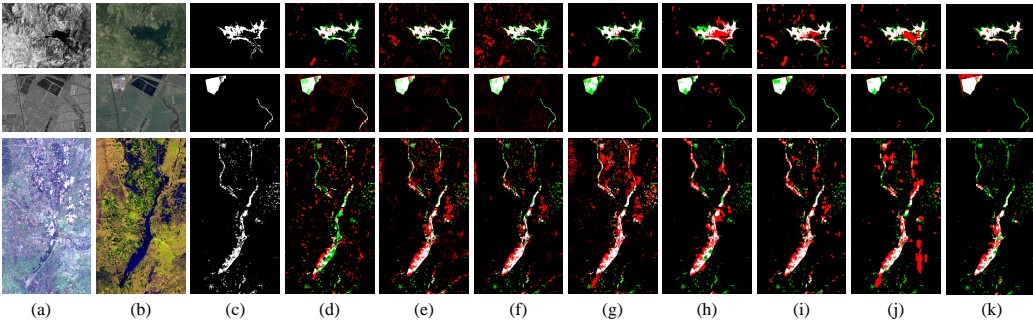

(a)     (b)     (c)     (d)     (e)     (f)     (g)     (h)     (i)     (j)     (k)

Figure 3: Experimental results of different methods over three datasets. From top to bottom are the Sardinia, Shuguang, and California datasets, respectively. (a) Pre-event image $X$; (b) Post-event image $Y$; (c) Ground truth; (d) CCLMRF; (e) X-Net; (f) ACE-Net; (g) PMBCN; (h) NPSG; (i) INLPG (j) SRGCAE; (k) SBGC. (The changed and unchanged parts are shown in white and black, whereas FP and FN are shown in red and green)

Italy, including a near-infrared (NIR) image and a three-band multispectral image acquired in 1995 and 1996 using Landsat-5 and Google Earth sensors. These images are sized at $300 \times 412$ pixels and depict the changes resulting from lake expansion; (2) the Shuguang dataset (Liu et al., 2018) consists of a SAR image and a multispectral image, both $593 \times 921$ pixels, collected in 2008 and 2012, showing the changes of a piece of farmland in Shandong Province, China; and (3) the last dataset is the California dataset (Luppino et al., 2019), consisting of a pair of multispectral and SAR images taken before and after a flood in 2017, with a size of $875 \times 500$ pixels. Comprehensive detail regarding these datasets is summarized in Table 1.

## 3.2 IMPLEMENTATION DETAILS

In our experiments, the image patch size $p$ is set to 9. We utilize the SGD optimizer (Sutskever et al., 2013) with a learning rate of $5e^{-2}$ for network training. A batch size of 2048 is adopted, and the number of epochs is set to 50. The proposed SBGC is compared with seven unsupervised SOTA methods: CCLMRF (Mignotte, 2022), X-Net (Luppino et al., 2022), ACE-Net (Luppino et al., 2022), PMBCN (Liu et al., 2022), NPSG (Sun et al., 2021b), INLPG (Sun et al., 2022), and SRGCAE (Chen et al., 2022) to verify its superiority. Our experiments are conducted on a personal computer with Python 3.7.13, PyTorch 1.12.1, and an NVIDIA GeForce RTX 3090 GPU. To evaluate the performance of different CD methods, six common evaluation metrics are employed, including false negative (FN), false positive (FP), overall errors (OE), overall accuracy (OA), Kappa coefficient (KC), and computation time (CT). Then OE, OA and KC are defined as

$$OE = FP + FN \tag{9}$$

$$OA = \frac{TP + TN}{TP + TN + FP + FN} \tag{10}$$

$$PRE = \frac{(TP + FP)(TP + FN) + (TN + FN)(TN + FP)}{(TP + TN + FP + FN)^2} \tag{11}$$

$$KC = \frac{OA - PRE}{1 - PRE} \tag{12}$$

where TP and TN represent the values of true positives and true negatives, respectively. Lower values for FP, FN, OE, and CT, and higher values for OA and KC indicate better CD performance.

## 3.3 EXPERIMENTAL RESULTS

Our proposed SBGC outperforms seven existing methods on three multimodal datasets, achieving notable enhancements in terms of OA, KC, and CT metrics. The change maps and evaluation metrics for different methods across three datasets are presented in Fig. 3(d)-(k) and Table 2. The Sardinia dataset, despite its small size, contains many intricate details that challenge the network's ability to accurately detect changes. On this dataset, it is obviously observed that other methods shows a

Table 2: Change detection results of different methods on the three datasets. The best results are in bold, and the second-best results are underlined.

| Dataset | | CCLMRF | X-Net | ACE-Net | PMBCN | NPSG | INLPG | SRGCAE | SBGC |
|---|---|---|---|---|---|---|---|---|---|
| | FP | 2703 | 17188 | 8061 | _2332_ | 5702 | 7403 | 5518 | **1388** |
| | FN | 2716 | _1539_ | 2843 | 3146 | 2293 | **1494** | 2751 | 2147 |
| Sardinia | OE | _5419_ | 18727 | 10904 | 5478 | 7995 | 8897 | 8269 | **3535** |
| | OA(%) | _95.61_ | 84.85 | 91.18 | 95.57 | 93.53 | 92.81 | 93.31 | **97.14** |
| | KC(%) | _62.10_ | 33.19 | 42.26 | 59.71 | 53.78 | 54.35 | 50.59 | **74.09** |
| | CT(s) | 92.08 | 497.10 | 348.57 | 101.43 | 79.75 | **23.18** | 288.93 | _74.63_ |
| | FP | 26963 | 12795 | 16641 | **1350** | 12068 | 6635 | _5506_ | 11482 |
| | FN | 5314 | _4593_ | 5451 | 13004 | 4904 | 8226 | 7638 | **3482** |
| Shuguang | OE | 32277 | 17388 | 22092 | _14354_ | 16972 | 14861 | **13144** | 14964 |
| | OA(%) | 94.09 | 96.82 | 95.95 | _97.37_ | 96.89 | 97.28 | **97.59** | 97.26 |
| | KC(%) | 52.21 | 68.58 | 61.94 | 61.53 | 68.80 | 68.00 | _71.39_ | **72.86** |
| | CT(s) | 212.84 | 1211.23 | 1086.10 | 230.12 | _210.74_ | 237.64 | 332.72 | **145.51** |
| | FP | 20574 | 31133 | 22852 | 39232 | _17271_ | 21080 | 25662 | **7849** |
| | FN | 11960 | 6006 | 11457 | **4125** | 8756 | _6211_ | 8995 | 10542 |
| California | OE | 32534 | 37139 | 34309 | 43357 | _26027_ | 27291 | 34657 | **18391** |
| | OA(%) | 92.56 | 91.51 | 92.16 | 89.80 | _93.88_ | 93.58 | 91.85 | **95.68** |
| | KC(%) | 26.82 | 37.60 | 43.29 | 36.70 | 41.26 | _45.22_ | 33.01 | **46.12** |
| | CT(s) | 140.74 | 812.21 | 751.10 | 318.57 | 198.39 | 420.3 | _136.11_ | **124.73** |

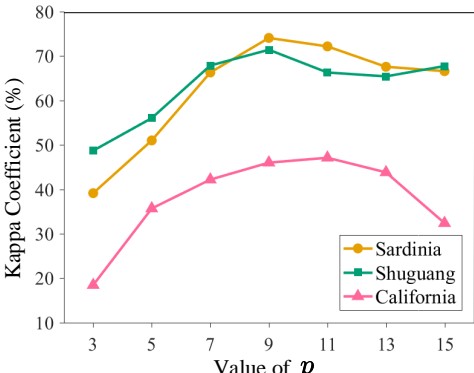

Figure 4: Relationship between the image patch size $p$ and Kappa Coefficient (KC) values of the proposed SBGC on diffrent datasets.

great amount of FP, misclassifying unchanged areas as changed (with large red areas ), along with relatively low OA and KC, indicating limited accuracy and reliability. In contrast, SBGC achieves superior performance by accurately distinguishing between unchanged and changed regions, detecting more truly changed regions. This is supported by Table 2, SBGC shows an improvement in KC of 40.90% (X-Net), 31.83% (ACE-Net), 14.38% (PMBCN), 20.31% (NPSG), 19.74% (INLPG) and 23.5% (SRGCAE). On the Shuguang and California datasets, which have more complex land cover information, the proposed SBGC also obtains accurate change maps with only a few noisy points, outperforming other comparative methods. Among these methods, INLPG and SRGCAE exhibit competitive performance, with SRGCAE achieving the second highest KC value on the Shuguang dataset, only 1.47% lower than our proposed SBGC. Although the KC values of SBGC are comparable to the second-best methods, it significantly reduces the time cost with the lowest CT, highlighting its practicality. Overall, the experimental results validate the effectiveness and practicality of the proposed SBGC in both quantitative and qualitative evaluations.

### 3.4 HYPERPARAMETER ANALYSIS

In our proposed SBGC, the size of the input image patch $p$ plays a key role in self-supervised feature learning, which affects the final performance of CD. Here, we provide an analysis of this parameter setting.

Table 3: Ablation studies of the proposed SBGC

| Components | | Dataset | | | | | |
|---|---|---|---|---|---|---|---|
| SSL | BGC | Sardinia | | Shuguang | | California | |
| | | OA | KC | OA | KC | OA | KC |
| ✗ | ✗ | 92.94 | 55.90 | 95.88 | 64.01 | 92.72 | 41.75 |
| ✔ | ✗ | 96.57 | 70.65 | 96.96 | 70.77 | 95.45 | 44.91 |
| ✔ | ✔ | **97.14** | **74.09** | **97.26** | **72.86** | **95.68** | **46.11** |

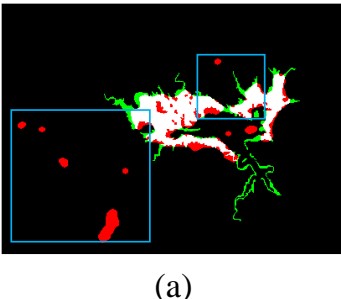 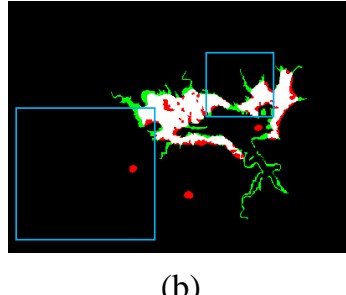

(a)  (b)

Figure 5: Visual comparison of the change maps on the Sardinia dataset. (a) without BGC (b) with BGC.

To detect changes in land cover more accurately, we set a smaller image patch size. Keeping other settings constant, we explore the influence of $p$ through experiments. Fig. 4 illustrates the variation of KC values with the increase of $p$. It can be seen that setting $p$ too large introduces redundant ground object information, whereas setting it too small fails to capture accurate object information, both of which lead to decreased CD performance. The KC value peaks at $p = 9$ and subsequently declines for $p > 9$. Notably, the best results are obtained with $p = 9$ for the Sardinia and Shuguang datasets, while $p = 11$ yields optimal results for the California dataset. For simplicity, we select $p = 9$, as it provides relatively good performance across all three datasets.

### 3.5 ABLATION STUDIES

To verify the validity of SSL and BGC, we conducted ablation experiments on three datasets. Two variants are designed for comparison: (1) a basic framework without SSL or BGC, where the original image information replaces the features learned by SSL, and BGC is substituted with a one-way graph comparison; (2) a framework that incorporates SSL without BGC; and (3) our complete method with SSL and BGC.

Table 3 shows the contribution of each component in our method across three datasets. The utilization of features extracted through SSL outperforms the use of original image information, highlighting the ability of SSL to provide more representative and robust features. Moreover, incorporating BGC enhances the OA and KC values across all datasets. This demonstrates that compared to one-way graph comparison, BGC can capture more comprehensive change information and reduce false alarms, thus improving CD performance. This is further supported by Fig. 5. As shown in Fig. 5(a), the change map obtained without using BGC has more false alarms and does not sufficiently extract change information. In contrast, our complete method with BGC improves the detection of changed areas and reduces false alarms, as depicted in Fig. 5(b). When SSLN and BGC are used together, the best performance can be observed (the third row of Table 3). Overall, the effectiveness of the proposed method can be demonstrated through the analysis provided above.

## 4 CONCLUSION

This paper proposes a novel self-supervised network, SBGC, for heterogeneous CD in remote sensing, addressing the critical challenge of detecting changes in ground objects across heterogeneous images. Initially, we employ pseudo-Siamese networks to extract representative and robust features

from original image regions based on SSL, thereby accurately reflecting complex land cover information. Additionally, BGC is introduced to effectively explore the change information within the graph structures. The remarkable experimental results on three heterogeneous datasets have validated the superiority and robustness of the proposed method over seven existing unsupervised SOTA methods. In future work, we intend to design more powerful self-supervised learning tasks to acquire more representative features, thereby improving the performance of heterogeneous CD in more complex scenarios.

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
