# OpenReview forum: "SBGC: Bidirectional Graph Comparison-Based Self-Supervised Network for Change Detection in Heterogeneous Images"
_ICLR.cc/2025/Conference — Submitted to ICLR 2025_

### Official Review · Reviewer_QBj9 · 2024-10-29

**Soundness:** 3
**Presentation:** 3
**Contribution:** 3
**Rating:** 5
**Confidence:** 4

**Summary:**

This paper addresses the challenge of performing change detection on heterogeneous images, particularly in the context of limited labeled samples. It proposes a self-supervised network called SBGC. The main contributions of SBGC are as follows:
1. By applying data augmentation to unlabeled samples and using contrastive learning to train the encoder, the model achieves more robust feature representation extraction.
2. This paper introduces Bidirectional Graph Comparison (BGC), which enables comprehensive extraction of rich information from graph structures.
3. Experiments conducted on three datasets demonstrate the superiority of SBGC.

**Strengths:**

1. The novel application of a self-supervised learning approach, which effectively circumvents the need for large labeled datasets. This approach is not only innovative but also practical for real-world remote sensing applications where labeled samples are scarce.
2. Its innovative use of Bidirectional Graph Convolution (BGC) to address the limitations of existing methods that rely on one-way difference calculations for feature extraction. By implementing BGC, it achieves bidirectional feature extraction, leading to comprehensive and accurate capture of graph information.
3. Superior accuracy compared to existing self-supervised methods, showcasing its potential to advance the state-of-the-art in heterogeneous change detection.

**Weaknesses:**

1. The final difference features are produced using a direct averaging operation, which is a relatively simple and traditional approach. To make this more effective, consider exploring alternative approaches for combining difference features, such as weighted averaging or learned fusion methods. These approaches could potentially improve performance by allowing more adaptive and context-sensitive feature integration.
2. The final segmentation results rely on Otsu’s thresholding method, a widely used but earlier technique. The choice of this method is not well justified in the paper, and no comparisons with more modern approaches, such as DeepLab [1] and SegFormer [2], are provided to demonstrate its effectiveness. Including such experiments would strengthen the validation of the proposed method and offer a clearer understanding of its advantages.

[1] Chen L C, Papandreou G, Kokkinos I, et al. Deeplab: Semantic image segmentation with deep convolutional nets, atrous convolution, and fully connected crfs[J]. IEEE transactions on pattern analysis and machine intelligence, 2017, 40(4): 834-848.

[2] Xie E, Wang W, Yu Z, et al. SegFormer: Simple and efficient design for semantic segmentation with transformers[J]. Advances in neural information processing systems, 2021, 34: 12077-12090.

**Questions:**

1. To provide a more comprehensive evaluation, the paper would benefit from the inclusion of additional performance metrics such as F1-score, Intersection over Union (IoU). F1-score would be particularly informative as it balances precision and recall, offering insight into the model’s accuracy and robustness. IoU, on the other hand, is especially useful for assessing segmentation-based change detection, as it provides a direct measure of overlap between predicted and actual change regions.
2. In this paper, the decision to use a direct averaging operation for generating the final change map needs further justification. It is essential to explain why this straightforward approach was chosen and how it contributes to the overall performance of the model.
3. Additionally, the use of Otsu’s method, a simple threshold segmentation algorithm, should be supported with clear reasoning. To enhance the credibility of the chosen approach, the authors should provide experimental results comparing Otsu’s method with other algorithms.

---

### Official Review · Reviewer_MoKw · 2024-10-31

**Soundness:** 3
**Presentation:** 2
**Contribution:** 2
**Rating:** 5
**Confidence:** 4

**Summary:**

This paper resolved an unsupervised heterogeneous image change detection problem using a graph-based approach.
They first used self-supervised learning (contrastive learning) to train two modality-specific encoders.
Using well-trained encoders, they extracted deep features from multi-modality remote sensing images and used the similarity of deep features to construct KNN graphs for consequent bidirectional graph comparison.
OTSU algorithm was finally used for thresholding and yielding the binary change map.
The experiments were conducted on three multi-modal image pairs, where the image sizes were around <1,000 x <1,000 pixels.

**Strengths:**

1. The idea of bidirectional comparison is promising and makes sense.
2. The proposed method achieved the best accuracy and fastest inference speed compared to seven conventional methods on three image pairs.

**Weaknesses:**

1. The motivation for the usage of the graph is unclear. The whole framework follows self-supervised representation and then thresholding. Graph stuff is mainly used to aggregate related spatial context. It is hard to understand why authors introduce the graph stuff here. The necessity of the graph should be carefully clarified. For theoretical insights or something else?

2. The major difference between the proposed method and the conventional method is the bidirectional comparison. Why do you want to do it bidirectionally? The motivation is barely stated in the paper (why the previous method didn't use that information, e.g., line 228)

3. The right of Figure 1 is too messy to read.

4. please carefully check your formulas between lines 222 and 237, e.g., Y has two styles, which makes it difficult for readers to understand.

5. The evaluation datasets are too small, just three image pairs. More large-scale experiments are needed. Besides, in this case, the accuracy is more susceptible to variations due to randomness, so reporting the mean and variance of your accuracy is a good solution.

6. line 299, $5e^{-2}$ -> 5e-2

**Questions:**

1. Table 3, if you did not use SSL, how to compute deep features?
2. It seems like SSL contributes most to the proposed method. How do different SSL methods impact the final performance?

---

### Official Review · Reviewer_tLkC · 2024-11-01

**Soundness:** 3
**Presentation:** 2
**Contribution:** 2
**Rating:** 5
**Confidence:** 5

**Summary:**

The authors introduce a straightforward self-supervised method for heterogeneous image change detection. This approach focuses on learning representative features for patches of original images through self-supervised learning and constructs KNN graph based on patch features. For these KNN graphs, the proposed BGC perceives bidirectional changes to extract difference information and fuses them to obtain the output.

**Strengths:**

-  The proposed use method bested on image self-similarity. This can effective solve the problem that heterogeneous images are more difficult.
- Patch features are extracted through contrastive learning. This unsupervised method can effectively convert patches into representative feature vectors.
- BGC can deeply explore the information in graph structure. The bidirectional comparison method can effectively provide the difference information before and after the change. This method improves the defects of the one-way comparison in previous studies.

**Weaknesses:**

- For each modality, separate contrastive learning is used to extract patch features. I think it will lead to distribution discrepancy for features at different modality. When calculate graph mapping, Unaligned modals may lead to bias for $G_{x}^{\mathcal{Y}}, G_{y}^{\mathcal{X}}$. Please answer my questions from the following two aspects:
1. Could you explain the motivation for not doing modal alignment in the SSL part?
2. Please add an additional ablation experiment to verify that the existing construct is optimal. Specifically, the positive samples for contrastive learning should be selected from different augmentations of the same patch in the pre-event image and the post-event image. All other aspects should remain unchanged. The configuration for the ablation experiment should be similar to that in Table 3. Of course, if you have other methods for modality alignment, you can also add them to this ablation experiment.

**Questions:**

- The study by Sun et al is similar to your approach, which could lower the novelty of this paper. The authors are suggested to discussion the similarity and difference between INLPG[1].

[1] Sun, Yuli, Lin Lei, Xiao Li, Xiang Tan, and Gangyao Kuang. "Structure consistency-based graph for unsupervised change detection with homogeneous and heterogeneous remote sensing images." IEEE transactions on geoscience and remote sensing 60 (2021): 1-21.

---

### Official Review · Reviewer_qUPa · 2024-11-01

**Soundness:** 2
**Presentation:** 2
**Contribution:** 2
**Rating:** 3
**Confidence:** 4

**Summary:**

This paper proposes a self-supervised network based on bidirectional graph comparison (SBGC) for unsupervised heterogeneous CD, which exploits modality-independent structural relationships. The proposed method achieves impressive experimental results on three public heterogeneous datasets in comparison with seven state-of-the-art (SOTA) unsupervised heterogeneous CD methods, which validates the effectiveness of the model.

**Strengths:**

1. The bidirectional mechanism effectively utilizes graph structures from both image modalities, potentially reducing false detections and enhancing accuracy.
2. The use of self-supervised learning (SSL) with pseudo-Siamese networks for feature extraction is a robust approach that aligns well with the self-supervised learning trend.

**Weaknesses:**

1. The Otsu threshold segmentation plays a crucial role in generating the change map. A sensitivity analysis on the threshold parameter T would provide more insights into its effect on the final results.
2. The authors only state that SSL can extract better features, but they do not compare this with traditional non-self-supervised methods (e.g., networks trained without SSL) in terms of feature quality. A comparison based on feature representation quality (such as classification accuracy or clustering performance) would more strongly support the superiority of SSL in feature extraction.
3. Although the authors assert that SSL can extract more representative features, they do not include feature visualizations (such as t-SNE or UMAP) to show the difference in feature distribution before and after SSL. Providing visualizations of feature space could more clearly illustrate the improvements SSL brings to feature representation, making the claim more convincing.
4. The BGC is a central innovation of the method, but the ablation experiment only evaluates it by completely removing the entire module. This approach overlooks the impact of each stage within BGC, such as the difference between one-way and bidirectional comparisons or the specific mapping strategies in graph construction. By gradually removing or altering parts of the BGC (e.g., using only one-way comparison or changing the graph mapping method), the paper could provide a more nuanced understanding of how each element contributes to the model's overall effectiveness.

**Questions:**

please see the weaknesses

---

### Meta-Review · Area_Chair_HhvP · 2024-12-20

**Metareview:**

The authors propose a method for heterogeneous change detection that constructs a modality-independent graph representation to capture structural information in multi-modal images. These graph representations are then combined with Otsu thresholding for change detection. Experimental results demonstrate that the proposed method performs well across various datasets. Instead of relying on raw image features, the authors utilize contrastive learning to extract patch-level features.

While the reviewers acknowledge that the use of contrastive learning for patch-level feature extraction is an interesting approach and that the method achieves impressive results on the datasets presented, they express a major concern regarding its similarity to the works of Yuli Sun et al. Specifically, the proposed method employs patch-level feature extraction using contrastive learning, which is an incremental improvement over Yuli Sun et al.'s work. Notably, the authors did not address the reviewers' concerns about the novelty of their approach.

**Additional Comments On Reviewer Discussion:**

In addition to concerns about the lack of novelty in the proposed method compared to the works of Yuli Sun et al., the reviewers pointed out several other issues, including lack of motivations for algorithmic choices, shortcomings in the experiments, validity of ablation tests, and inadequate comparisons with other methods. However, the authors failed to address any of these concerns during the rebuttal phase.

---

### Decision · Program_Chairs · 2025-01-22

Reject